

# Effects of soil moisture and surface heat fluxes on the South American Monsoon System over West-Central Brazil: an observational study

João Pedro Nobre[1], Manoel Gan[1], and Éder Paulo Vendrasco[1]

[1]National Institute for Space Research (INPE), São José dos Campos, São Paulo, Brazil

**Correspondence:** João Pedro Nobre (joao.nobre@inpe.br)

**Abstract.** This study evaluated the relationship between different surface hydrometeorological variables and rainfall during the wet period of the South American Monsoon System (SAMS). A climatological study was performed for 30 rainy periods of the SAMS between the years 1991-2021 over the Central-West region of Brazil (WCB) (20-10° S and 60-50° W). The European Centre for Medium-Range Weather Forecasts Reanalysis 5th/ERA5 was used to understand, under different soil
moist conditions (dry, intermediate, and wet), the hydrometeorological patterns and their impacts on SAMS during the three stages of the wet period: the development (September-October-November, SON), maturity (December-January-February, DJF), and weakening (March-April-May, MAM) South American monsoon quarters. The results show that along with an increase (decrease) in rainfall during the rainy season, there is also a significant increase (decrease) in both surface and subsurface volumetric soil moisture ($\theta$) for the wet (dry) soil condition periods. However, the surface heat flux composites showed that
the latent heat fluxes to the atmosphere ($H_l$) significantly exceeds the climatology during the SAMS development quarters (SON) for the wet soil group. In contrast, for the dry soil group, the significant increase of $H_l$, compared to the climatology, over the WCB occurred only during the SAMS maturity quarters (DJF), representing, in this last case, a significant injection of latent heat and consequently delayed evaporation ($EVP$) compared to rainy periods with wet soil, due to the low soil moisture content and the prevalence of dry convection over WCB. Regarding the sensible heat fluxes ($H_s$), it was observed drier (wetter)
soils tend to exhibit values above (below) the climatological mean throughout all stages of SAMS evolution over the WCB. The 2-m air temperature ($T2m$) and planetary boundary layer height ($PBLH$) anomaly composites showed that for the wet (dry) soil group and with significant rainfall above (below) the climatological mean over the WCB, all evolution quarters were marked by a significant decrease (increase) in $T2m$ and $PBLH$ anomalies. Regarding the mechanisms of direct feedback between surface variables and rainfall associated with the SAMS, significant direct correlations were also observed between
the mean rainfall of the SAMS rainy season and the mean values of active days duration ($D_{ad}$), $H_l$, $H_s$ and the surface Soil Moisture Condition Index ($SMCI_1$).

## 1 Introduction

Climatologically, the rainy season associated with the South American Monsoon System (SAMS) begins in mid-September over the western Amazon basin, in mid-October and November, respectively, in Central-West and Southeast of Brazil, while





the rainy season demise occurs in early April in Central-West of Brazil (WCB) and in the second half of May over southern Amazon region (Silva and Kousky, 2012).

Furthermore, understanding the hydrometeorological patterns associated with SAMS (Kousky, 1988; Sansigolo, 1989; Marengo et al, 2001; Liebmann and Marengo, 2001; Nogués-Paegle et al, 2002; Gan et al, 2004, 2006; Garcia and Kayano, 2009, 2013; Santos and Garcia, 2016), as well as their relations with onset and demise of the rainy season, is crucial for strate-

gic planning across various society sectors. A shorter and drier rainy season, for instance, can lead to increased electricity costs, which has implications for consumers who will incur additional expenses for regular consumption. Additionally, for the success of crops with a short phenological cycle, such as soybeans and corn, particularly in the Southeast and Central-West regions of Brazil, a wet period with regular rainfall is essential.

Another relevant aspect for improving the understanding of SAMS rainfall is to comprehend the intraseasonal variability

of the rainy season in South America, which is associated with periods of above or below climatological normal precipitation (also known as active and break monsoon periods, respectively), largely linked to the presence or absence of the South Atlantic Convergence Zone (SACZ), as addressed by Nogués-Paegle and Mo (1997); Herdies et al (2002) and Ferreira and Gan (2011), and whose rainfall intensity is also influenced by aspects related to soil moisture and surface heat fluxes (Grimm et al, 2007; Grimm and Zilli, 2009).

Different authors (Chou and Neelin, 2001; Xue et al, 2006; Collini et al, 2008; Sörensson et al, 2010; Silva, 2012) also studied the relationship between rainfall with surface heat fluxes and soil moisture during the SAMS rainy period. The results of these studies concluded that the reduction of soil moisture in the SAMS rainy period contributes to decreasing the latent heat flux ($H_l$) and increasing the sensible heat flux ($H_s$), therefore, with a lower evaporation ($EVP$) rate and increase in the Bowen ratio ($B_o$), the volume of rainfall also decreases over areas of central Brazil.

In other regions across the globe, comparable outcomes to those attained by various authors regarding the South American continent to the soil moisture and surface heat fluxes have likewise been noted. For instance, Douville et al (2001) noted that increased soil moisture impacts the seasonal variability (March-September) of precipitation over the African continent. Likewise, various authors Zuo and Zhang (2007); Meng et al (2014) and Liu et al (2017) reported a significant positive correlation between soil moisture and summer rainfall in the context of the eastern Chinese monsoon. In the North American

Monsoon System region, Xue et al (2004) demonstrated, through a series of integrated numerical experiments, that precipitation increases in July when a prescribed anomaly of moist soil moisture is applied.

Although most studies consider the relationship between surface processes (surface heat fluxes and soil moisture) and rainfall during the rainy season of the South American Monsoon System (SAMS), there is a lack of studies that explore how these processes impact the onset and demise of SAMS rainy season, as well as the intensity and duration of the rainy season and

active and break periods. Additionally, gaining a better understanding of how different surface variables behave under varying conditions of surface and sub-surface soil moisture is another relevant aspect that requires further elucidation for WCB.

Thus, the aim of the present study is to analyze the impacts of surface heat fluxes and soil moisture on SAMS rainfall, their relationships with the lately and early onset and demise rainy season, as well as with the number and duration of active and break periods of rainfall during the wet period of the SAMS throughout the development (September-October-November,





SON), maturity (December-January-February, DJF), and weakening (March-April-May, MAM) South American monsoon

quarters with different soil moisture conditions (wet, intermediate and dry). In this regard, it was used the reanalysis data

from the European Center (European Center for Medium-Range Weather Forecasts Reanalysis 5th/ERA5) to understand the

behavior of different surface and subsurface hydrometeorological variables taking into account different soil moist conditions.

In Sect. 2 we describe the datasets and analyses procedure, methodology and criteria used to identify the onset and demise

dates of the rainy season, and the break and active periods. In Sect. 3 we present the results of the climatology of some

hydrometeorological variables, composite fields and a discussion about the rainfall and meteorological variables feedback

mechanisms during SAMS wet season. Sect. 4 summarizes the results and presents the conclusions.

## 2 Materials and Methods

### 2.1 The region

The area over WCB (20-10° S and 60-50° W, shown in Fig. 1) is used in the present study as a reference to provide a detailed

view of the impacts of soil moisture and surface heat fluxes in South America in the SAMS rainfall. This region is considered as

the core of the SAMS and, it is influenced by the main atmospheric circulation patterns of the South American monsoon rainy

season (Gan et al, 2004), in addition to covering a part of the maximum summer precipitation where the annual atmospheric

circulation cycle is highly related with SAMS.

The socioeconomic relevance of this region also justifies the area delimited for the present study, as it encompasses arable

areas, which together with other Brazilian regions contribute in 21% to the gross national product (Embrapa, 2020) and whose

crop success is highly dependent on a humid period with regular rainfall. Furthermore, this study area covers the headwa-

ters of the main South American basins, Amazonas and Prata, important for energy generation within the Brazilian National

Interconnected System (SIN).

### 2.2 Analyses

The data used in this study is the ERA5 reanalysis (Hersbach, 2023). This dataset combines model data with observations

from around the world through a globally complete dataset. It is available with a horizontal resolution of 0.25° latitude×0.25°

longitude, except for ocean variables, whose horizontal resolution is 0.5° latitude×0.5° longitude at hourly intervals from 1979

to the present for 37 pressure levels, ranging from 1000 to 1 hPa. Therefore, the ERA5 in this study represented the surface

and atmospheric climatological aspects over the South America Continent for the variables: precipitation (mm.day$^{-1}$), $H_l$ and

$H_s$ (W.m$^{-2}$), planetary boundary layer height ($PBLH$, m), volumteric soil moisture between 0-7 cm and 100- 289 cm ($\theta_{0-7}$

and $\theta_{100-289}$, respectively, in m$^3$.m$^{-3}$), and air temperature at 2 m ($T2m$, in °C).

The ERA5 variables were used to create composites of different hydrometeorological variables anomalies to understand the

climatological behavior of rainfall, $H_l$, $H_s$, $PBLH$ $\theta_{0-7}$, $\theta_{100-289}$, and $T2m$, as well as their relationships with three soil

conditions: dry, intermediate and wet, during the September-October-November (SON), December-January-February (DJF)



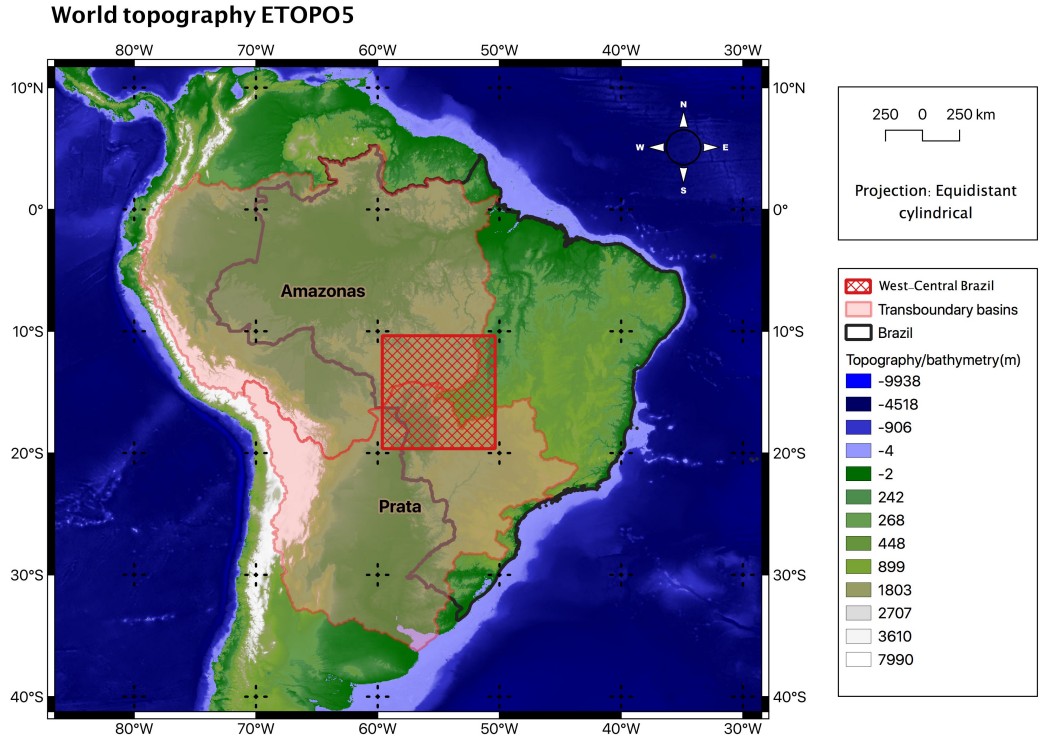

**Figure 1.** Digital terrain elevation map for the South American continent prepared using the Earth topography five-minute grid (ETOPO5, 2023) database. The red hatched square represents the area of WCB, the main study area in this work.

and March-April-May (MAM) quarters. These quarters comprise the developmental (SON), maturity (DJF), and weakening (MAM) stages of the rainy season associated with SAMS in WCB.

To obtain the quarterly composites, anomalies of various surface hydrometeorological variables were calculated, and the Student t-test (Student, 1908) was applied to the sample data to examine the null hypothesis that the mean of a sample drawn from a population is centered on a pre-specified value. Additionally, the student t test was employed in this study due to its extensive usage in climatological studies and easy interpretation, as well as its flexibility in distribution (e.g., Bombardi and Carvalho, 2017; Fialho et al, 2023).

The Pearson Correlation Coefficient (Freedman et al, 2007) was used to better understand the feedback patterns between surface and subsurface hydrometeorological variables mean ($H_l$, $H_s$, $\theta_{0-7}$, $\theta_{100-289}$ and $T2m$), $PBLH$ for all WCB wet season time duration, the onset and demise (early/late) dates, and the number of active/break periods with wet season rain mean over WCB.





The significance test for the Pearson Correlation Coefficient was calculated as proposed by Pitman (1937), Eq. 1. This method involves approximating the sample distribution of $r$ with a $\beta$-distribution and can be applied by consulting tables of the t-distribution.

$$t_c = \frac{r\sqrt{n-2}}{\sqrt{1-r^2}} \tag{1}$$

Where $n$ represents the number of data.

## 2.3    Clustering Algorithm

This study employed $\theta_{0-7}$ [Eq. (2)] to classify the soil into three groups: dry, intermediate, and wet. These classifications were crucial for assessing surface effects in the atmosphere because superficial soil moisture exhibits a strong interaction with rainfall variability when both variables are coupled, as discussed by Backer et al (2021); Dai et al (2022).

$$\theta_{0-7} = \frac{\theta_w}{\theta_s} \tag{2}$$

Where the volumetric soil moisture $\theta_{0-7}$ [Eq. (2)] is the ratio of the water volume in the soil sample ($\theta_w$) and the total volume of dry soil, air, and water in the soil sample ($\theta_s$).

     To determine the dry, intermediate, and wet soil moisture conditions, the Soil Moisture Condition Index [$SMCI$, Eq. (3), Zhang and Jia (2013)] was calculated for an average area over WCB between 20-10° S and 60-50° W from ERA5 daily data during SAMS wet season. The SMCI ranges from 0 for drier up to 100% for wetter surface soil condition. In a second step, we

calculated the $SMCI$ average per rainy season between 1991-2021, and, finally, the k-means clustering technique was applied to classify the average $SMCI$ sample by rainy season between 1991-2021 into three groups; dry, intermediate, and wet [for details about the applied k-means clustering algorithm, refers to Sect. 2.2 of Grønlund et al (2017)].

$$SMCI = 100 \cdot \frac{SSM_i - SSM_{min}}{SSM_{max} - SSM_{min}} \tag{3}$$

     In Eq. (3), $SSM_i$ is the $\theta_{0-7}$ daily average, $SSM_{max}$ and $SSM_{min}$ are, respectively, the maximum and minimum $\theta_{0-7}$ daily average compared to a 30-year dataset (1991-2021), whose database only has the days that make up the SAMS rainy

period. The $SMCI$ final results is a daily mean for all SAMS rainy period in a specific year, as showed in column $SCMI_1$ in Table 1.

## 2.4    The onset and demise

To determine the SAMS rainy season onset and demise, we used a similar methodology proposed by Gan et al (2004). In that methodology, the rainy season onset (demise) occurred when the first occurrence of 850-hPa westerly (easterly) winds along

60° W in the band 20–10° S together with rainfall rates were superior (less) than 4 mm.day$^{-1}$ for at least 6 of 8 subsequent





pentads over WCB between 20-10° S and 60-50° W. We did not use the zonal wind persistence criteria because a previous analyses showed the ERA5 zonal wind had high variability during the study period. Furthermore, as the rainfall database used in this study was not the same as the one used by Gan et al (2004), we recalculated the daily average rainfall for the WCB area, and the results were approximately similar to those of Gan et al (2004). Therefore, the threshold of 4 mm.day$^{-1}$ was retained
in the methodology of this study.

### 2.5 Active and break periods

The definition of active and break SAMS periods were carried out with Monsoon Precipitation Index (MPI), developed by Krishnamurthy and Shukla (2000) for the Indian monsoon region, and applied by Ferreira and Gan (2011) on WCB (20-10° S and 60-50° W). This index is directly related to the precipitation anomaly.

The MPI uses the daily precipitation climatology for the study area to obtain precipitation anomalies. The criterion used to identify the active (break) periods was the same used by Krishnamurthy and Shukla (2000) for India. Therefore, an active (break) monsoon periods occur when the MPI is a half standard deviation above (below) the average precipitation daily for at least five consecutive days. According to Ferreira and Gan (2011), the five-day threshold is applied to remove high-frequency variability associated with the passage of transient systems.

## 3    Results and Discussion

### 3.1    Climatology

During the SAMS development stage (SON), the rainfall peaks around the WCB region, with volumes greater than 4 mm.day$^{-1}$. This increase in rainfall over WCB is due to deep convection that initially increases over areas of the western Amazon basin in September and subsequently expands to areas of Central-West and Southeastern Brazil in November (e.g., Gan et al, 2004;
Silva and Kousky, 2012). This is a result of the maximum surface heating that occurs in these regions during the SON quarter.

In the SAMS maturity stage (DJF), rainfall over WCB can exceed 8 mm.day$^{-1}$ and it is connected with the Atlantic Intertropical Convergence Zone (ITCZ), and these values decline to 1-2 mm.day$^{-1}$ in the following quarter (MAM), when convection shifts towards the equator, and rainfall tends to be more restricted and voluminous over areas in the northwestern Amazon basin, as described by Kousky (1988) and Horel et al (1989).

Along with the gradual increase in rainfall over the WCB during the development and maturity stages of the SAMS, it is possible to observe an increase in $H_l$ and volumetric soil moisture between 0-7 cm (Fig. 3 and Fig. 2, respectively) compared to the JJA quarter (period of SAMS inactivity). In other words, in parallel with the increase in evaporative demand and surface soil moisture over the WCB, there is also an increase in the intensity of rainfall over the WCB during the SON and DJF quarters ([Fig. 3]). However, when the precipitation weakens (during the withdrawal quarter of the SAMS), the volumetric soil moisture
between 0-7 cm (Fig. 2) and $H_l$(Fig. 3) reach values on average comparable to those of the development quarter of the SAMS,





representing a decrease pattern compared to the previous quarter (DJF), with values respectively equivalent to 0.400-0.425 $m^3.m^{-3}$ (Fig. 2) and 80-100 $W.m^{-2}$ (Fig. 3).

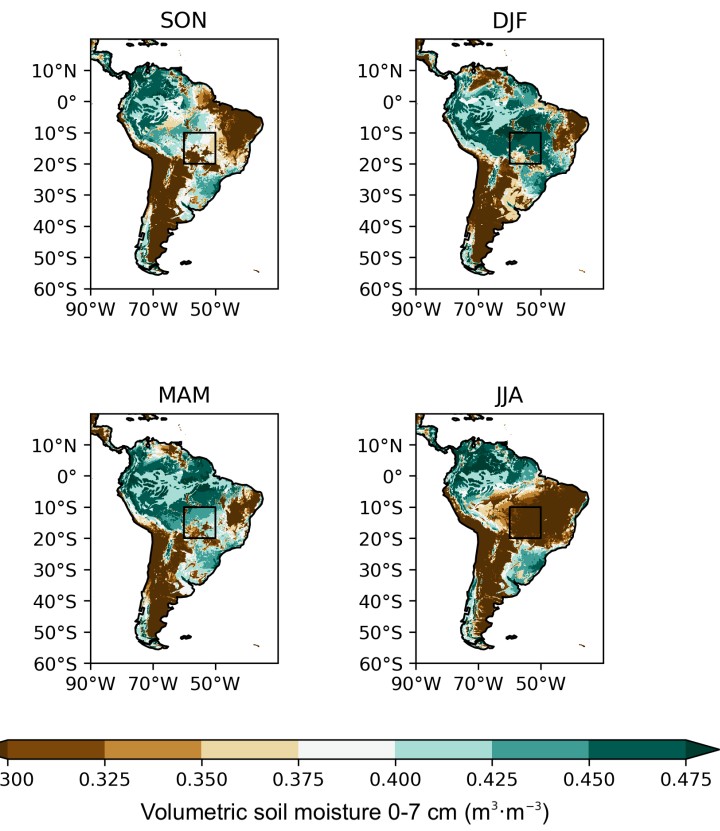

**Figure 2.** Climatological map of volumetric soil moisture 0-7 cm ($m^3.m^{-3}$) over the South American continent during the September-October-November (SON), December-January-February (DJF), March-April-May (MAM), and June-July-August (JJA) quarters. The black square demarcates the WCB region.

Regarding the $H_s$, it is possible to observe (Fig. 3) lower values during the SAMS maturity quarter between 20-40 $W.m^{-2}$, when compared to the development quarter and these values extended until the weakening stage. In addition, during the development quarter, the values of the $H_s$ play, in general, values relatively comparable to those obtained for winter months in South America (values around 40-60 $W.m^{-2}$) and higher than those verified in the SAMS maturity and weakening stages.

The relative increase of $H_s$ along with $T2m$, with values greater than 30 °C, during the SAMS development quarters over the WCB demonstrates that the fraction of radiation balance directed to surface heating is a crucial factor for atmospheric instability. On the other hand, during the SAMS maturity and weakening quarters, values around 25 °C are observed over the WCB, which are relatively comparable to those obtained during the winter quarter (JJA) in the same area.



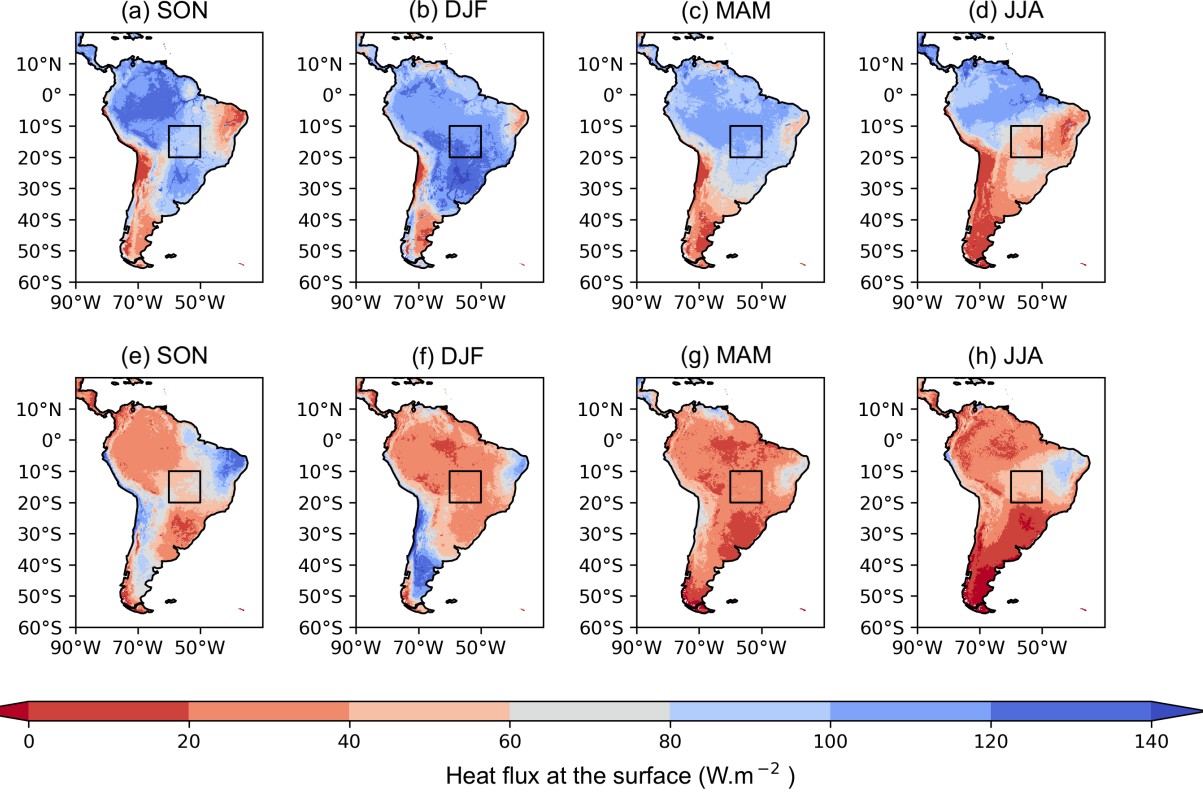

**Figure 3.** Similar to Fig. 2, but for latent heat flux at the surface (a-d) and sensible heat flux at the surface (e-h) in W.m$^{-2}$.

Collini et al (2008) also reveals similar aspects to those found in the climatological fields of this study for the surface heat fluxes and soil moisture, however, here is discussed to all stages of the SAMS rainy season evolution. About the relationship between near-surface air temperature and rainfall, Grimm et al (2007) also show, similarly to what was found in the climatological field of this study, that there is a positive relationship between surface temperature during spring and the amount of rainfall that occurs in Central-East Brazil during summer.

Therefore, during developmental and maturity SAMS stages, South America receives higher solar radiation due to its position relative to the Sun. This leads to increased heating of the land surface, promoting higher $EVP$ rates and convective instability, consequently a greater release of latent heat. Furthermore, during these two stages (developmental and maturity SAMS stages) the South America are also characterized by increased soil moisture availability, which is influenced by the convergence of moisture fluxes from the Amazon basin and the Subtropical Atlantic Ocean, that contribute to increased rainfall and soil wetting over the WCB. Thus, the gradual increase in soil moisture together the solar radiation activity promote a gradual $EVP$ rates and latent heat fluxes at the surface increase during developmental and maturity SAMS stages. In contrast, during weakening SAMS stage, the incidence of solar radiation gradually decreases together the SACZ actuation, resulting





in reduced $EVP$ rates, latent heat flux and soil moisture available at the surface, which contributes to reducing the rainfall
180    volumes over the WCB.

However, considering the climatology previously presented, some doubts still persist in the the existing literature; what would be the behavior of these different hydrometeorological variables under conditions of dry, intermediate, and wet soils? Do they have any relationship to the recurring rainfall during SAMS rainy period? The answers to these questions can be found in the following section.

185    ### 3.2    Climate feedbacks

The one-dimensional clustering method k-means allowed the separation of soil moisture conditions into three groups: dry, intermediate, and wet; whose data values regarding $SMCI_1$ used to create the groups can be found in Table 1. Moreover, from this Table 20% of the wet periods are part of the driest soil group between 1991-2021, corresponding to the rainy seasons over WCB with years marked with $\triangledown$. The group with intermediate soil moisture conditions presented the highest amount of data, 190    totaling 50% of the evaluated wet periods, and includes the rainy seasons marked with $\lozenge$. The group with wet soil conditions encompasses 30% of the evaluated wet periods, which include the rainy seasons marked with $\triangle$.

The creation of these three groups throughout the rainy season over WCB (which typically spans from October to April) was essential to examine the relationship between surface soil moisture and different hydrometeorological variables. Moreover, this approach facilitated a more in-depth understanding of how these variables behave under distinct soil conditions, in accordance 195    with a more recent climatology (1991-2021), across the stages of development, maturity, and weakening of the rainy season. It is important to note that much of the existing literature predominantly focuses on the development or maturity stages of the rainy season period, lacking a refined analysis of the prevailing soil moisture condition throughout the entire rainy season.

#### 3.2.1    Soil moisture and rainfall

The Fig. 4 shows significant $\theta_{0-7}$ anomalies during the mature stage of the rainy season (DJF) for dry soil group (exhibiting 200    negative $\theta_{0-7}$ anomalies over the southwest region of the WCB) and wet soil group (displaying positive $\theta_{0-7}$ anomalies over the WCB southwest region).

During the quarters corresponding to the development stage (SON) and weakening stage (MAM), positive $\theta_{0-7}$ anomalies were also significant over the study area to wet soil, as showed in Fig. 4. However, in the dry soil group, areas with significant negative $\theta_{0-7}$ anomalies were concentrated more in the southwest portion of the WCB.

205    In development and maturity stages of the rainy season, $\theta_{0-7}$ anomalies showed values below the climatological mean for the intermediate soil, but without statistical significance for the study area (Fig. 4). However, during the weakening stage, significant negative $\theta_{0-7}$ anomalies are observed for the group with intermediate soil (Fig. 4). This result is an indicative that along with the typical decrease in rainfall intensity during the weakening quarter (MAM), there is also a reduction in $\theta_{0-7}$ over WCB.



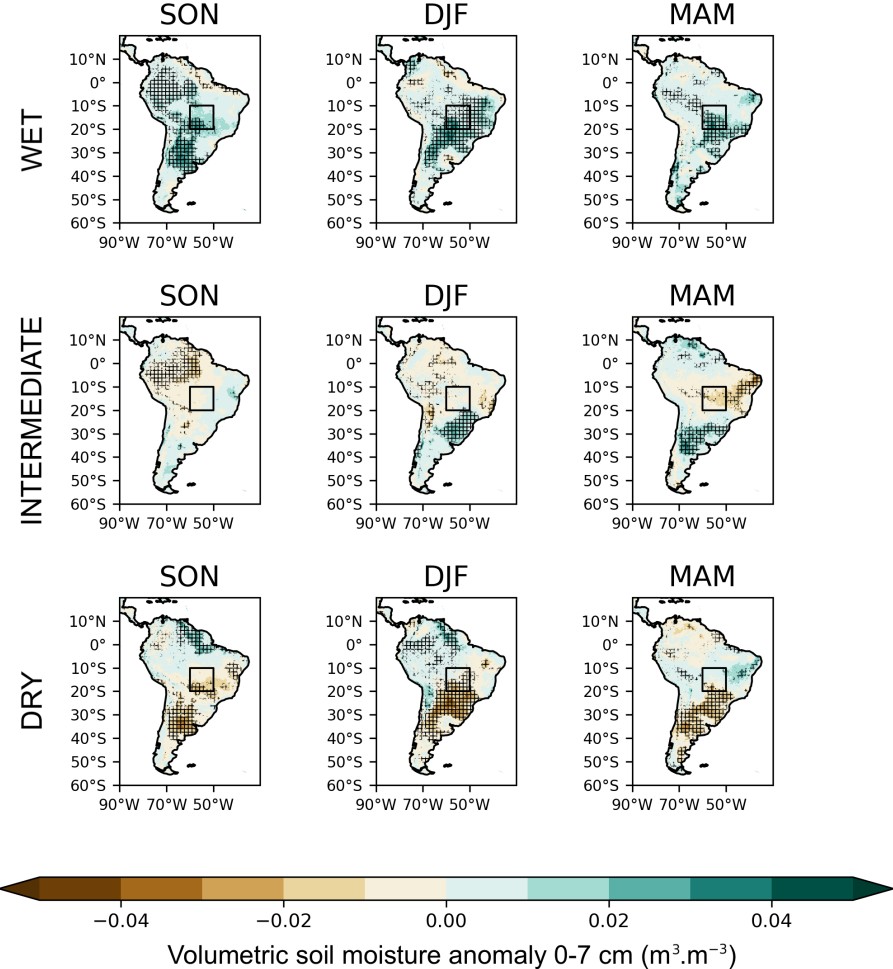

**Figure 4.** Quarterly composite data for September-October-November (SON), December-January-February (DJF), March-April-May (MAM) for volumetric soil moisture anomalies at a depth of 0-7 cm. The top row shows the quarterly composites for rainy seasons with wet soils, the second row shows the quarterly composites for rainy seasons with intermediate soil conditions, and the third row, from top to bottom, shows the quarterly composites for rainy seasons with dry soils. These composites were obtained from the monthly ERA5 dataset between 1991-2021. The hatched area shows statistical significance results for p-values less than 0.05.

Similar to what was identified for the $\theta_{0-7}$ anomalies, for volumetric soil moisture between 100-289 cm ($\theta_{100-289}$), Fig. 5, it was also possible to observe positive soil moisture anomalies throughout all stages of the rainy season for the group with wet soil, which may support a longer duration of active periods of the rainy season, as described in Fig. 7 of this study.

In the case of the dry soil group (Fig. 5) significant negative $\theta_{100-289}$ anomalies are observed during the maturity (DJF) and weakening stages of the rainy season in almost all WCB. However, for the quarter corresponding to the development stage, significant negative soil moisture anomalies were concentrated in the southwest portion of the study area. For the cases with



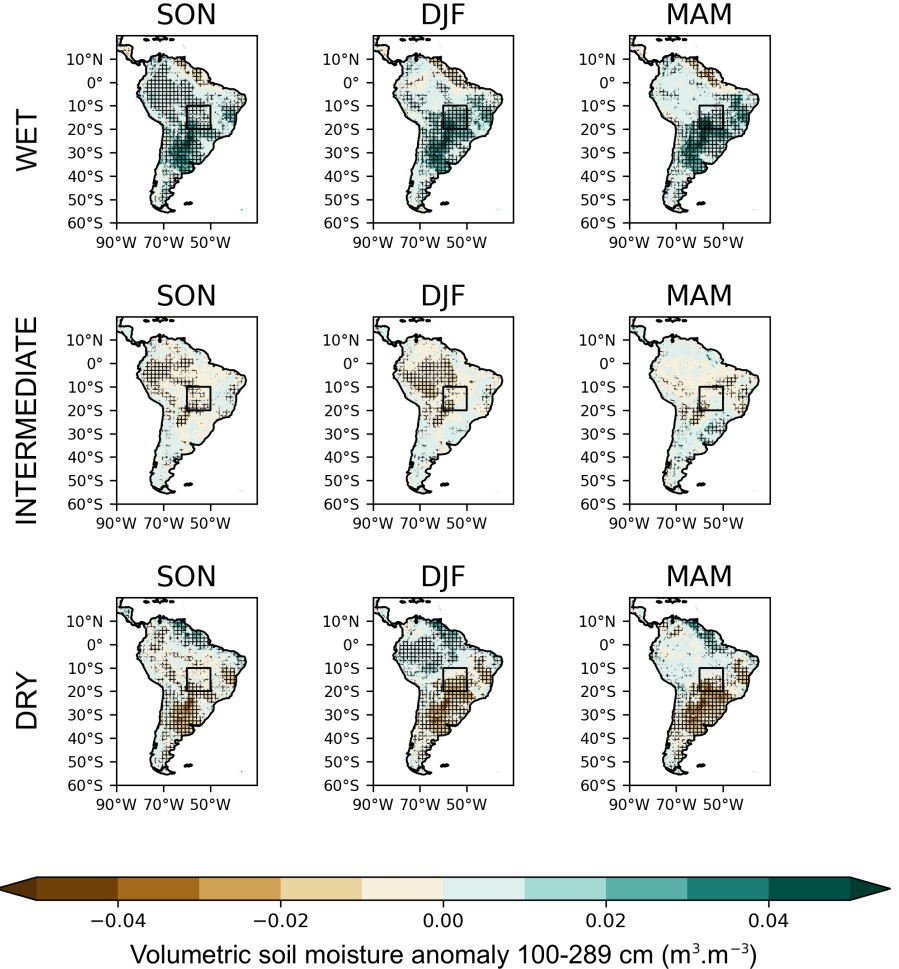

**Figure 5.** Similar to Fig. 4, but for volumetric soil moisture between 100-289 cm.

intermediate soil, significant negative $\theta_{100-289}$ anomalies were observed, concentrated in the southwest portion of the WCB, indicating a drier pattern compared to climatology.

The Fig. 6 shows the rainfall anomaly composites created for each of the groups with dry, intermediate and wet soils during the SAMS evolution stages. This figure shows that there are positive precipitation anomalies in the WCB in all stages of the SAMs for the wet soil group, as observed in the anomaly fields of $\theta_{0-7}$ and $\theta_{100-289}$ for all SAMS evolution stages. Significant negative anomalies of rainfall were more notable in the western (SON) and northeastern (MAM) portions of the WCB for the intermediate soil group, as well as in the southern portion of the WCB during the SAMS development and weakening rainy season stage for the dry soil group. For the group of intermediate soils, predominant positive precipitation anomalies are observed during SAMS mature stage over the WCB region, although without statistical significance.





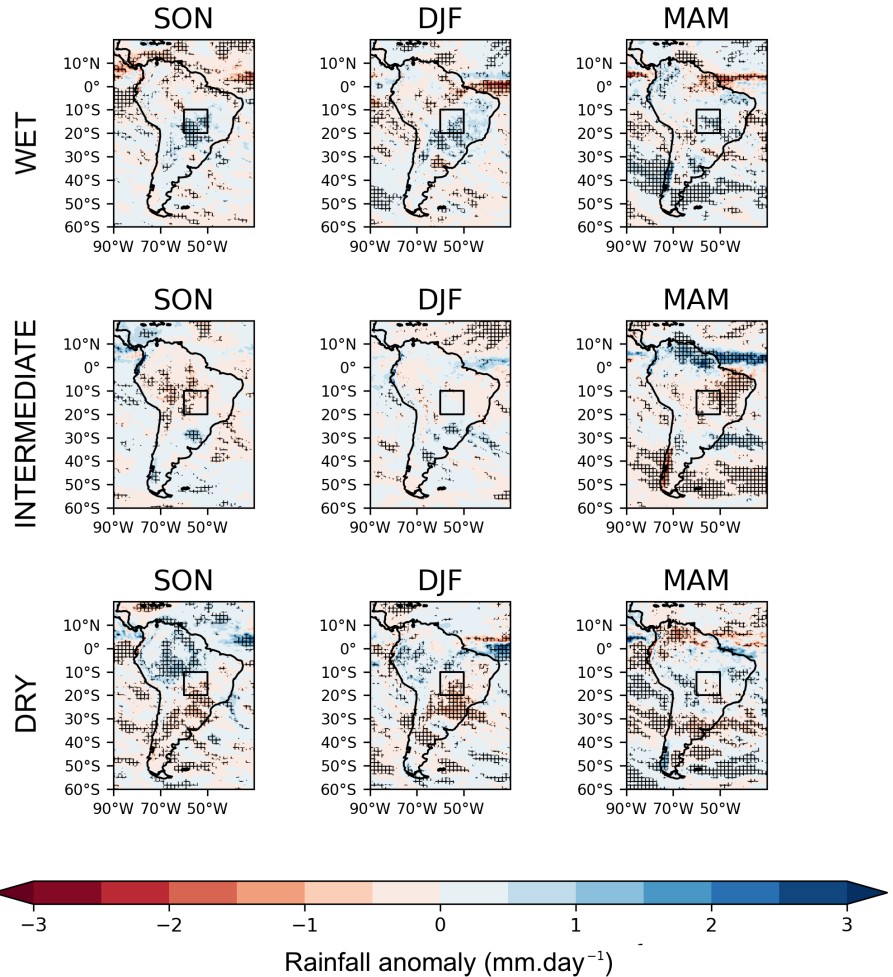

**Figure 6.** Similar to Fig. 4, but for rainfall anomaly.

These results presented in this section agree with those obtained by Eltahir (1998), Collini et al (2008), Sörensson et al (2010), and Sörensson and Menéndez (2011) who observed that under wetter soil conditions, rainfall increases over areas in WCB during the SAMS rainy season, while for relatively drier surface soils, areas with below-average rainfall become more significant, mainly in in the south and southwest WCB areas. Similar findings to those obtained in this study were also observed in other regions influenced by a monsoon precipitation in African continent Douville et al (2001), China (Zuo and Zhang, 2007; Meng et al, 2014; Liu et al, 2017) and North America Xue et al (2004).

Thus, as illustrated in Fig. 7, there is a statistically significant positive correlation (70%) between rainfall and $SMCI_1$ during the wet season of SAMS. This imply a proportional relationship indicating, on average, the wetter (drier) the season, correspond to higher (lower) soil moisture in superficial layers. In this same perspective, studies like Bedoya-Soto et al (2018), Backer et al (2021), and Dai et al (2022) also show that soil moisture is a key variable in defining spatiotemporal precipitation





patterns, as moist soil can act as a reservoir that provides moisture to the atmosphere, giving an extra support of humidity for cloud and rain formation.

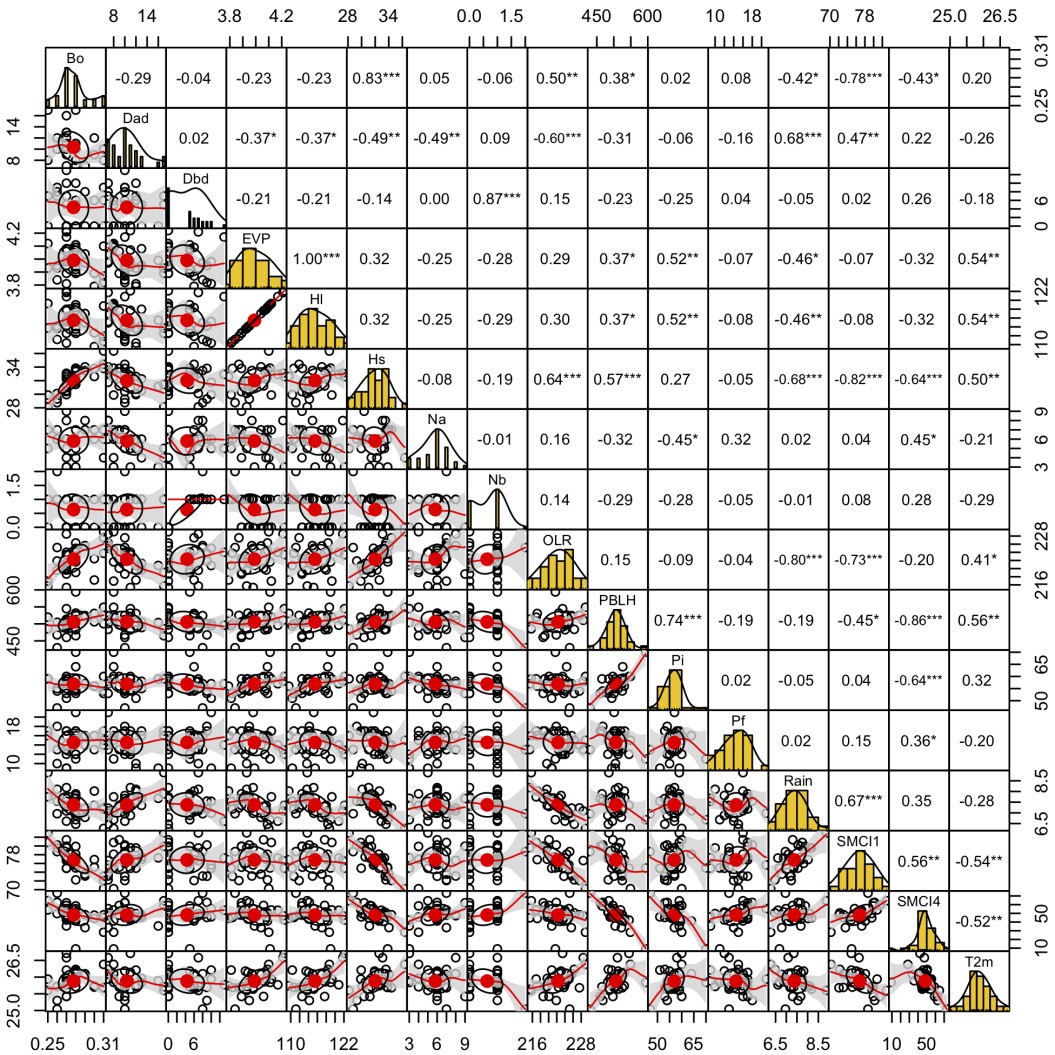

**Figure 7.** Correlation matrix to assess the degree of relationship between mean rainfall for the rainy season period with $B_o$, $H_l$, $H_s$, $SMCI_1$, $SMCI_4$, $T2m$, $PBLH$, onset ($P_i$) and demise ($P_f$) pentads of the rainy season, and number and duration of active and break periods of the SAMS rainy season. On the main diagonal, frequency histograms for each evaluated variable were plotted. Values to the right of the main diagonal represent the degree of interrelationship between all analyzed variables, considering the statistical significance results for p-values less than 0.001 (***), 0.01 (**), 0.05 (*), 0.1 (.), 1(). On the left of the main diagonal, scatter plots for all analyzed variables are displayed, with the red dot representing the centroid of each scatter plot.



The reason for the increase in precipitation over areas of the WCB during the rainy season of SAMS under wetter soil conditions is primarily due to enhanced evaporation ($EVP$) and convective activity. When the soil is wet, there is a greater availability of moisture near the surface, which promotes higher rates of $EVP$. This increased moisture supply leads to higher
atmospheric humidity levels, providing more fuel for convection and cloud formation. As a result, more intense and frequent rainfall events occur, leading to an overall increase in precipitation over the WCB.

Conversely, for relatively drier surface soils, areas with below-average precipitation become more significant. When the soil is dry, there is limited moisture available for evaporation and convective processes. As a result, atmospheric humidity levels may be lower, leading to reduced cloud formation and less intense convection. This can result in decreased rainfall and a higher
likelihood of experiencing below-average precipitation anomalies over WCB regions.

However, a secondary situation to what is commonly observed in the literature can also be encountered during wet periods when the actual vapor pressure over the WCB region is high and the soils are relatively wetter. This situation involves a decrease in evaporation rates during SAMS wet season. This occurs because once the soil is saturated and the actual vapor pressure (which can be influenced, for example, in a given year when moisture advection is more effective over the WCB
region) is also high, the atmosphere can require a lower evaporative demand compared to relatively drier soils (with lower actual vapor pressure).

Another result, alike to the findings of Sörensson et al (2010), the intensity of rainfall over the Atlantic ITCZ during the SAMS wet season exhibits a strong positive correlation with the presence of wetter soils across areas of the South American continent, including WCB. Therefore, the maps in Fig. 6 reveal that during the quarter of maximum development of the SAMS
rainy season, under wet surface soil conditions over WCB, the region with the highest precipitation volume near the equator (around $0°$, where the ITCZ is located) experiences the lowest rainfall volumes further south and positive anomalies further north compared to the 1991-2021 climatology. Conversely, under dry soil conditions over WCB, the pattern is reversed. This pattern can mean that the ITCZ is positioned further north when in wetter soil conditions and further south when the soil is drier.

### 3.2.2 Temperature and rainfall

In the composite of $T2m$ (Fig. 8) it can be observed that the group of rainy seasons with wet soil showed significant negative anomalies during all stages of SAMS development over WCB and neighborhood regions. This phenomenon occurs due to the prevalence of convective activity, especially during the development (SON) and maturity (DJF) stages of SAMS for wet soil groups, which contributes to the establishment of a potentially wet and unstable environment. This environment favors the
development of convective clouds over many regions of South America, which, combined with large-scale circulation, leads to less efficient surface radiative heating, primarily caused by the presence of dense and persistent clouds when compared to the dry soil group.

Regarding the $T2m$ composites for the group with intermediate and dry soil conditions, there is a predominance of significant positive anomalies of $T2m$ over the WCB. In the intermediate soil group, they prevail in most of the evaluated area during DJF
and in the northeast during MAM. Already for the dry soil group, positive $T2m$ anomalies prevail in most of the evaluated area



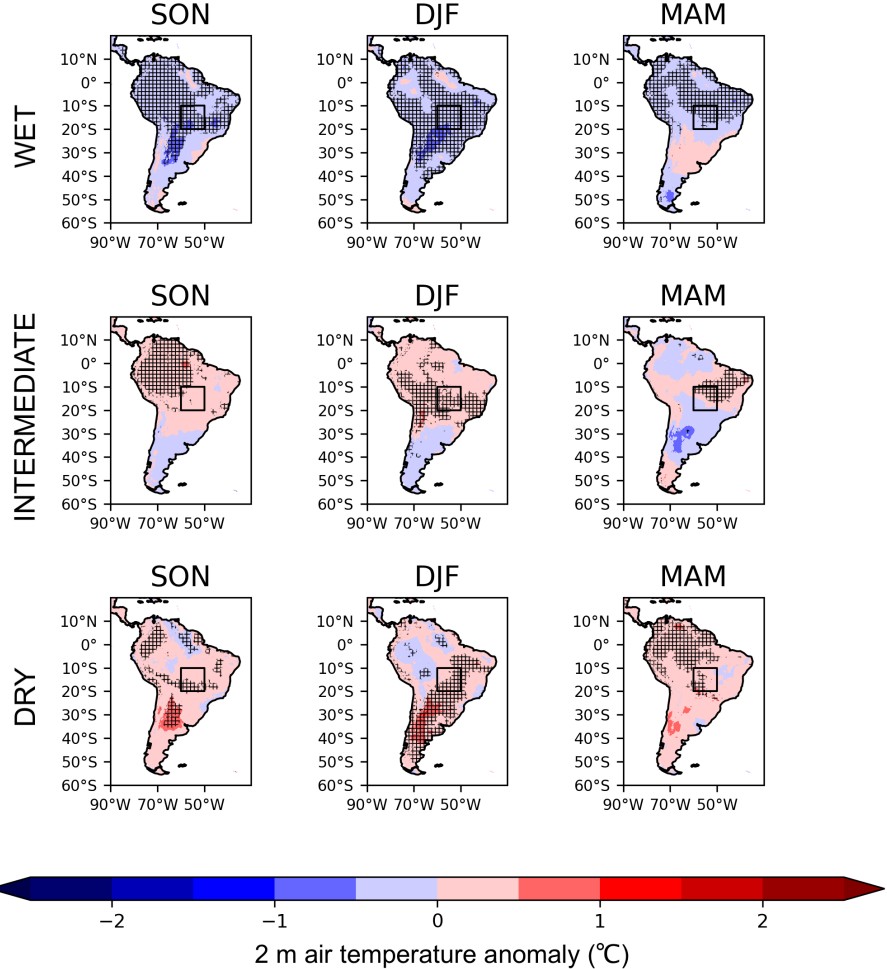

**Figure 8.** Similar to Fig. 4, but for 2-m air temperature anomaly.

during DJF and in the southwest sector of the WCB during MAM. This phenomenon occurs because the radiation that reaches the surface can promote more effective surface heating compared to the group of moist soils, due the presence of less dense and persistent clouds over the WCB.

Therefore, the findings suggest that during the rainy season, above-normal precipitation is more strongly influenced by a positive anomaly in soil moisture rather than a positive anomaly in temperature. Although warmer air can result from various factors such as solar heating, advection of warm air, and adiabatic heating due to downward motion, the availability of sufficient moisture becomes crucial for the production of increased precipitation.

An interesting pattern can also be observed when comparing the $PBLH$ (Fig. 9) and $T2m$ (Fig. 8) fields. In rainy seasons with wet soils, a predominance of negative anomalies is observed over areas in the WCB for $PBLH$ throughout all stages of SAMS evolution, where negative anomalies of $T2m$ are also evident. However, for intermediate and dry soils, positive



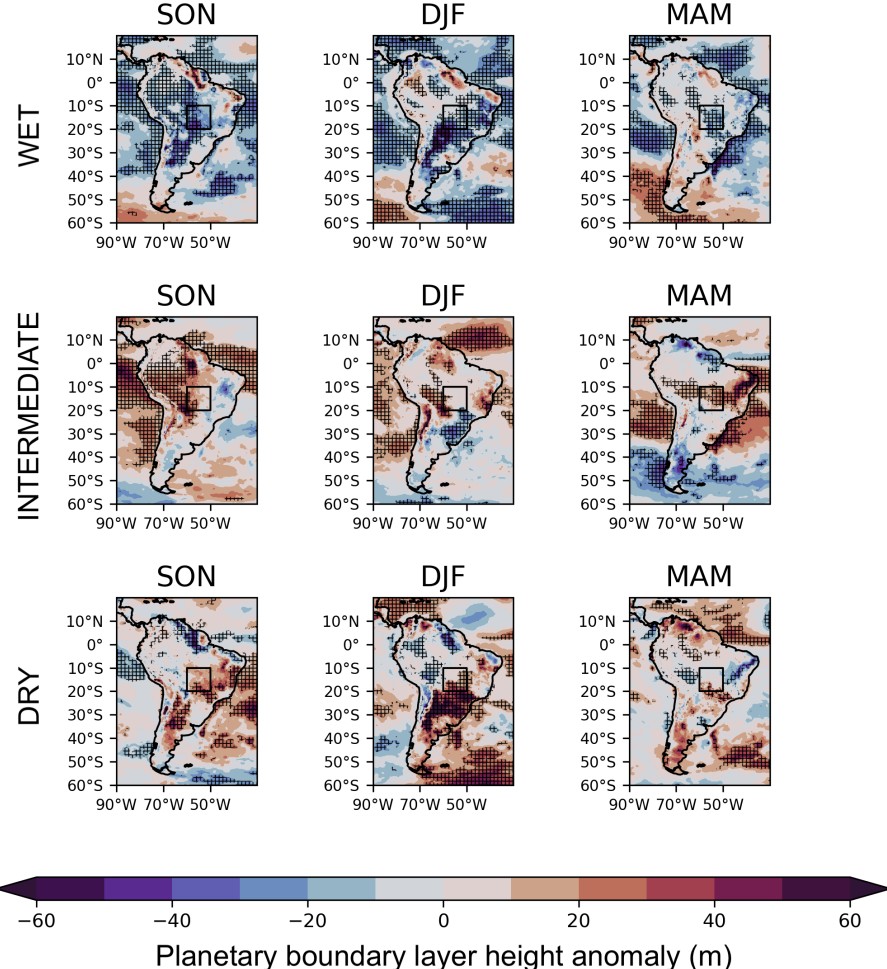

**Figure 9.** Similar to Fig. 4, but for planetary boundary layer height.

anomalies of $PBLH$ prevail, consistent with areas where significant positive anomalies of $T2m$ are present. This phenomenon arises because turbulence is caused by the vertical movement of air and warm air is more likely to move vertically than cold air. This increases the turbulence in the boundary layer and therefore increases its height.

While the mean $T2m$ and $PBLH$ during the SAMS rainy season do not exhibit direct and significant correlations with the rainfall over WCB, as depicted in Fig. 7, an indirect relationship can be discerned between both variables and rainfall. Hence, in situations where there is an anomalous wetness (dryness) of the soil, deep (shallow) convective clouds coupled with a higher (lower) $D_{ad}$ values contribute to an ineffective (effective) surface solar heating. Consequently, this leads to a reduced (elevated) $T2m$ and $PBLH$ anomalies, because dense (shallow) and persistent (brief) clouds typically recurrent in SAMS rainy season with wet (dry) soil contribute to reduce (increase) the solar radiation heating at the surface. This shows that the combination of

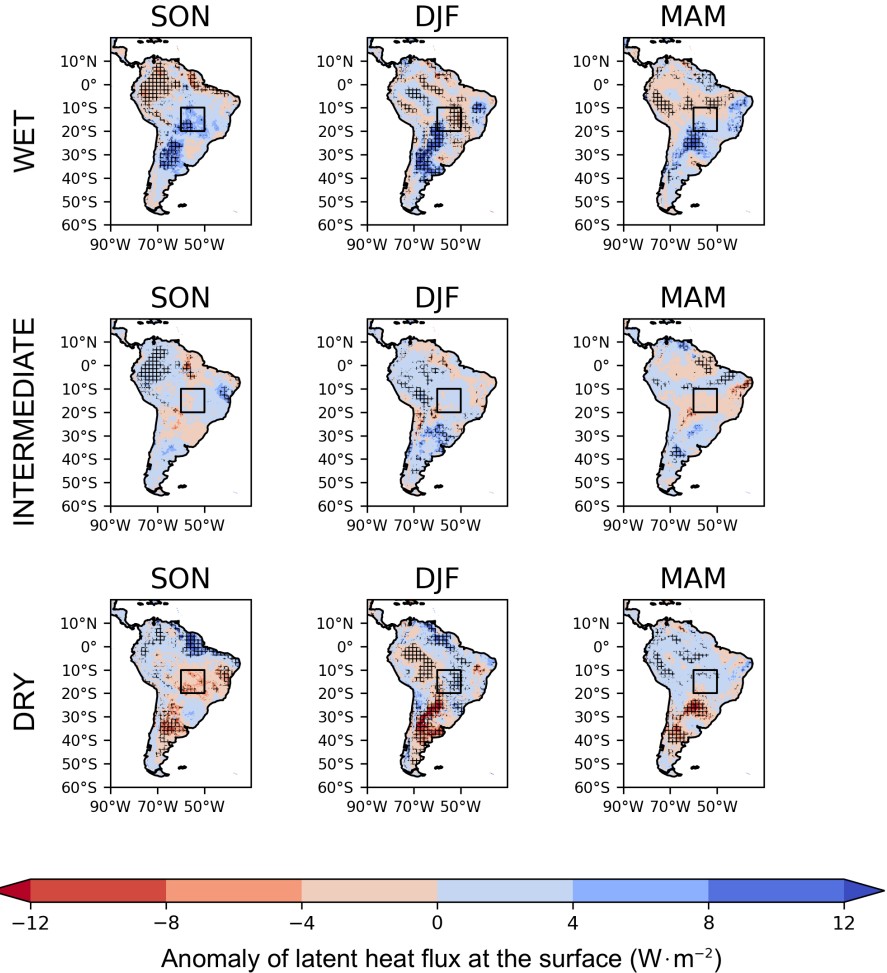

**Figure 10.** Similar to Fig. 4, but for latent heat flux anomaly.

surface processes along with convection driven by the dynamic component of the atmosphere are essential elements in shaping
a SMAS rainy season characterized by more or less rainfall.

### 3.2.3   Surface heat fluxes and rainfall

The quarterly composite for $H_l$ (Fig. 10) also provided significant opposing patterns for the SAMS development (SON) and
maturity (DJF) stages for wet and dry soils. Fig. 10 shows during the development stage, the group with wet soil had significant

$H_l$ positive anomalies over the evaluation stage, associated, therefore, with a higher injection of $H_l$ when compared with the
dry soil composites, where a significant decrease of $H_l$ is noted compared to the climatology over the southwest portion of the
WCB.





In the SAMS maturity quarter for the wet soil group (Fig. 10), a significant decrease in $H_l$, concentrated mainly in the WCB eastern portion, and a persistent increase over the southwest portion was observed. Conversely, in the dry soil group, a

significant increase in $H_l$ can be seen over WCB eastern region and a persistent decrease over the southwest region.

During the SAMS weakening stage, $H_l$ positive anomalies are observed over the WCB southwest portion for the group with wet soil condition (Fig. 10), while negative anomalies are observed for this same region of the WCB in the dry soil group, reinforcing surface heat fluxes opposite patterns in all stages of SAMS evolution for the $H_l$ of wet and dry soil groups.

As for the $H_l$ patterns for intermediate soil conditions, values above the climatological average for the WCB are observed in

the SAMS development and maturity quarters (Fig. 10). In the weakening quarter (Fig. 10), there is a predominance of negative anomalies for $H_l$. However, in all stages of SAMS development, $H_l$ anomalies for the intermediate soil group did not present statistical significance.

The anomaly composite fields of $H_s$ demonstrate a consistent prevalence of significant negative anomalies throughout all stages of SAMS evolution for the wet soil group (Fig. 3). Conversely, for the dry soil group, there is a persistent predominance

of significant positive anomalies of $H_s$ over the WCB across all stages of SAMS, indicating that the fraction of the radiation balance allocated for surface heating is much higher for the rainy season group with dry soil compared to the wet soil group.

For the intermediate soil group (Fig. 11), a predominance of positive $H_s$ anomalies can be observed over WCB during the SAMS development and maturity quarters, but they did not show statistical significance. Additionally, for this same soil group, there is a significant increase in $H_s$ anomalies during the SAMS weakening quarter.

Beyond the study area located in WCB, there is also a curious heat flux pattern in the composites of dry and wet soil groups for the area near 30° S and 65° W, as shown in Fig. 10 and 11, respectively. For the wet soil group, characterized by above-average rainfall in WCB (Fig. 6) during all stages of SAMS development, a significant increase in $H_l$ anomalies is observed in all quarters evaluated in this area (Fig. 10). On the other hand, for the dry soil group, when below-average rainfall is observed in WCB, there is a significant decrease in $H_l$ anomalies (Fig. 10). For $H_s$, a contrary pattern is observed over the region around

30° S and 65° W; there is a decrease in $H_s$ for the wet soil groups and an increase in $H_s$ in the dry soil groups during all stages of SAMS evolution, as showed in Fig. 11. Similarly, the same region also exhibited persistent anomalous patterns throughout the entire SAMS rainy period, behaving in an antagonistic manner during years characterized by wet soil conditions (positive rainfall and surface/subsurface soil moisture anomalies, but negative $T2m$ and $PBLH$ anomalies) and dry soil conditions (negative rainfall and surface/subsurface soil moisture anomalies, but positive $T2m$ and $PBLH$ anomalies), as showed in Fig.

4, 5, 6, 8 and 9.

The relationship between the magnitude of rainfall in WCB and the regions near 30° S and 65° W was confirmed by Reboita et al (2010); however, the dynamic explanation for this phenomenon was well elucidated by Campetella and Vera (2002). They explain that the Low-Level Jet (LLJ) and cyclonic disturbances at the surface in the southern part of South America exhibit a feedback effect. In other words, the presence of a weak cyclonic disturbance in the southern part of South America enhances the

strengthening of the LLJ as it advances towards the La Plata basin region. Simultaneously, the jet stream, which transports heat and moisture, fosters the intensification of the cyclonic disturbance, thereby contributing to the enhancement and persistence





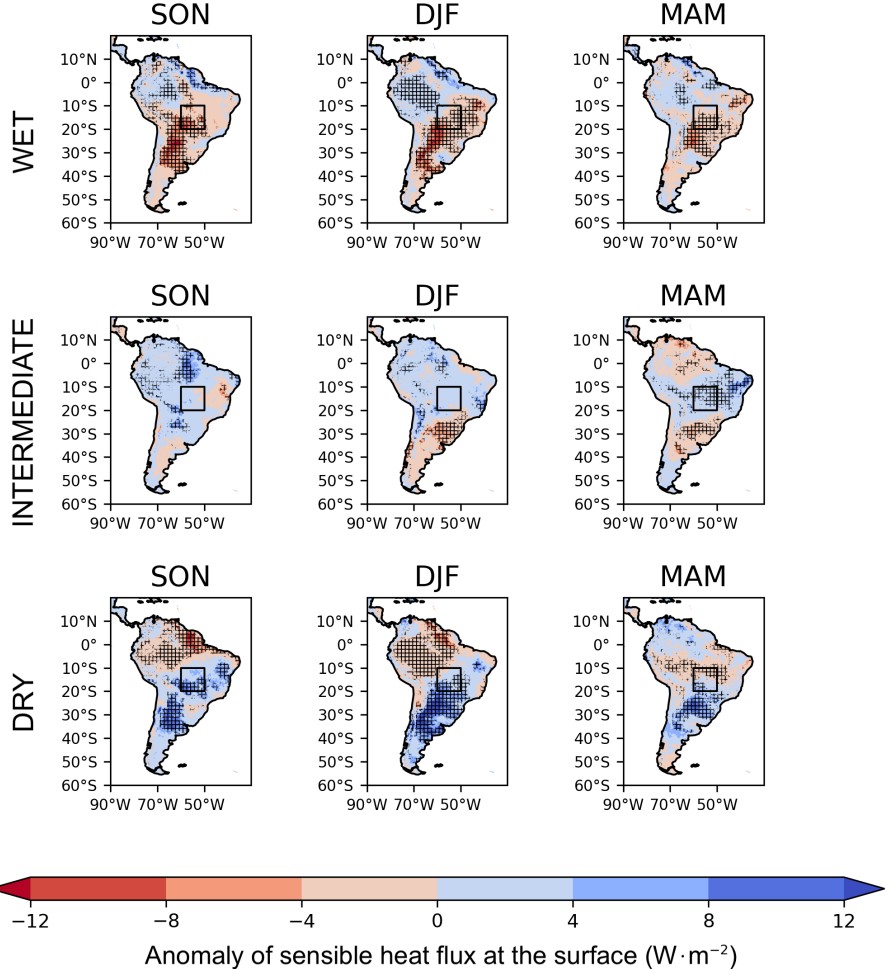

**Figure 11.** Similar to Fig. 4, but for sensible heat flux anomaly.

of the LLJ, soil wetting, and increased heat flux during the rainy season with wet soils. Conversely, for drier soils, an opposite scenario is observed.

As described by Fu and Li (2004), Xue et al (2006) and Neves et al (2013), and as demonstrated in the climatological

fields (Fig. 3), the SAMS rainy season is marked by a gradual significantly increase in $H_l$, especially, as showed in this study, in the SAMS development and mature stages of WCB wet and dry soil groups, respectively (Fig. 10). Regarding the $H_s$, it tends to increase during the development quarter and decrease during the maturity and weakening quarters over WCB (Fig. 3). However, as demonstrated in this study, when subjected to different soil moisture extreme conditions, i.e., dry and wet, $H_s$ tends to perform above and below the climatological average, respectively, throughout all stages of SAMS development (Fig.

340 11).





Thus, it is worth mentioning that the convergence of moisture fluxes from the Amazon basin and the Subtropical Atlantic Ocean (SACZ precursor element), as mentioned by Reboita et al (2010), followed by an increase in $\theta$ are essential factors during the onset and maintenance of the SAMS rainy season. In this perspective, it is believed that the moisture convergence from the Amazon basin and the Atlantic Ocean, serves as a mechanism, in the SMAS development stage, that increases the $\theta$,

but throughout the rainy season, surface-atmosphere feedback mechanisms act together to increase the SAMS rainfall volumes at the wet season (as observed in the climatological fields of Fig. 3 and 2).

As demonstrated in the climatological composites (Fig. 4, 5 and 10), the $\theta$ over WCB causes an anomalous increase (decrease) in mean daily precipitation, given the anomalous increase (decrease) of $H_l$ over WCB [i.e., more evaporation (minus evaporation)] in rainy periods with wetter soils, during SON (alread for drier soils, during DJF). Thus, for rainy periods with

drier soils, significant latent heat flux anomalies increase over the WCB are verified in a later quarter to the rainy periods with wetter soils, due to the low soil moisture content and the prevalence of dry convection over the study area.

It was also possible to observe in the $T2m$ climatological composites (Fig. 8), that in rainy periods with humid soils (dry) the decrease (increase) of the $T2m$ occurred over the WCB area demarcated by rainfall above (below) the climatology. Indicating that the presence of dense and persistent clouds over the WCB, when under wet soil conditions, allows a significantly anoma-

lous decrease of the $T2m$, while under drier soil conditions with less cloudness, $T2m$ anomalous heating was predominant over the WCB. In other words, compared to the climatology, the $H_l$ (Fig. 11) promotes more efficient surface heating during wet season with dry soils then wet soils.

The $H_s$ rainy season mean (Fig. 7), also showed statistically significant correlations at the 0.1% level with rainfall. As showed in Fig. 7, during periods of higher (lower) rainfall, $H_l$ and consequently $EVP$ are relatively lower (higher), indicating that

the amount of water evaporated from the soil into the atmosphere is relatively higher (lower) during periods of lower (higher) precipitation. It is important to note that these results may initially seem contradictory to those obtained from climatological composites, but this contrast can be explained by considering that relatively drier periods in this context also include rainy periods with dry and intermediate soil conditions.

Therefore, the significant negative correlation between $H_l$ and precipitation in WCB (Fig. 7) is related to the level of convec-

tive activity in the study area. The presence of fewer (more) days with deep convective clouds (as inferred from the correlation between precipitation and $D_{ad}$ in Fig. 7 and convection, figure not shown) during relatively drier (wetter) rainy periods contributes to a relatively higher (lower) amount of solar radiation reaching the Earth's surface, which contributes to a higher (lower) evaporation of content of water stored in the surface soil. This is especially evident during the SAMS wet seasons with intermediate soil conditions in WCB, where the surface soil layer is wetter compared to the SAMS with dry soils and the rain-

fall anomalies is close to the climatology along SMAS development stages of the rainy season over WCB. Given this aspect, note (Fig. 10) the prevalence of positive $H_l$ anomalies, although not statistically significant, over the WCB during the SAMS development and maturity quarters for intermediate soil group. Another important finding is that intermediate soil group has a lower actual vapor pressure than wet soil group and their soil is more wet compared to dry soil group, and this combined elements also contribute to intermediate soil group has a higher evaporating demand to the atmosphere compared with the other

soil groups.





Finally, The $B_o$ also exhibits a significant negative correlation (equivalent to 42%) with recurrent rainfall during the SAMS wet season. As shown in Fig. 7, the $B_o$ decreases (increases) when the volume of rainfall during the wet period of the SAMS increases (decreases), indicating the $H_l$ ($H_s$) is more significant than the $H_s$ ($H_l$) during wet periods marked by higher (less) volumes of rainfall, as also demonstrated by Xue et al (2006), Collini et al (2008), and Silva (2012).

### 3.3 Relationship with the onset and demise, number and duration of active and break days of the SAMS wet season with rainfall

The $D_{ad}$ (Fig. 7) also showed a positive correlation of 68%, indicating on average, the rainy periods with higher (lower) rainfall indices are also those with longer (shorter) periods of active days in WCB. For the remaining variables, $D_{bd}$, $N_a$, $N_b$, $P_i$ and $P_f$ no direct significant correlations with rainfall over WCB can be observed. While our focus has been on exploring the direct

relationships of these variables with rainfall, we must acknowledge that the brevity of this study does not preclude the existence of other mechanisms of indirect feedback between these variables and rainfall. Such indirect relationships will be a subject of investigation in future research.

As example, although no direct relationships were found between the mean volume of rainfall over WCB and $P_i$ and $P_f$ during the wet season of the SAMS, an interesting relationship was identified between $SMCI_4$ and the onset and demise

pentads of the South American monsoon season. As shown in Fig. 7, on average, $SMCI_4$ for the wet season of the SAMS with relatively wet (dry) soil, the onset of the rainy season occurs earlier (later), as well as a later (earlier) demise of the rainy season. This occurs because along the rainy season, the water content in the surface layers keeps relatively high and it is first consumed by $EVP$, which results in less variability to the deeps soil moisture.

Thus, as shown by Han and Zhoul (2013), soil drying by $EVP$ is often divided into two stages: during stage 1, $EVP$ occurs

at the soil surface and is limited by atmospheric evaporative demand. As drying progresses and surface moisture is depleted, the $EVP$ rate falls below the potential rate, and stage 2 or decreasing $EVP$ rate begins. During this stage, the location of $EVP$ shifts from the surface to the subsurface, resulting in the formation of a dry surface layer.Therefore, since the subsurface moist soil layer can only be accessed after the surface moisture reserves are depleted through $EVP$, it is expected that in extremely dry (wet) rainy periods, the subsurface soils remain predominantly dry (wet) throughout the entire SAMS wet season, due to

the scarcity (excess) of rainfall over the WCB to maintain the surface layer moist, thus showing a stronger relationship with the onset and demise of the SAMS rainy period.

## 4 Conclusions

The analysis of climatological fields has provided a better understanding of the behavior of different hydrometeorological variables at the surface during the stages of development, maturity, and weakening of the South American Monsoon System

(SAMS). In the monsoon development stage (SON), precipitation over West Central Brazil (WCB) is most concentrated in the northern and eastern areas of the region, with values between 7-8 mm/day, while in the southeastern region of the same quadrant, values between 5-6 mm/day are observed. The increase in precipitation over WCB in DJF is primarily attributed





to the presence of deep convection, initially intensifying over the western sector of the Amazon basin in September and subsequently spreading to Central-Western and Southeastern Brazil from November, as a result of maximum surface heating in these regions. During the maturity stage (DJF) of the SAMS, precipitation over WCB exceeds 8 mm/day, and in the following quarter (MAM), precipitation tends to be more concentrated and voluminous over the Amazon basin area, while decreasing over WCB.

An increase in $H_l$ and $\theta_{0-7}$, as observed in climatological fields, also contributed to the intensification of precipitation during the SON and DJF quarters. However, for $H_s$, its values were lower during the maturity stage of the SAMS. During the development stage, the $H_s$ flux played a crucial role in destabilizing the atmosphere. Additionally, it was possible to infer a more pronounced relationship between the rainiest quarter (DJF) of the SAMS and a relative decrease in $T2m$ over the WCB, compared to the SON and MAM quarters, because it is a quarter dominated by a greater persistence of convective activity over the region.

Through the anomaly composites obtained for different soil moisture conditions groups (dry, intermediate, and wet), a relationship between surface soil moisture and certain surface and subsurface hydrometeorological variables over WCB was observed. The significant positive and negative anomalies for $\theta$ were identified for both surface soil (0-7 cm) and subsurface soil (100-289 cm) in the wet and dry soil groups, respectively, during all stages of South American Monsoon System (SAMS) development, which coincided with positive (in the wet soil group) and negative (in the dry soil group) rainfall anomalies during the SAMS evolution quarters. For the intermediate soil group, a significant relationship with abnormally positive and/or negative values for surface and subsurface moisture in the WCB was generally not observed in any of the SAMS evolution quarters.

The latent ($H_l$) and sensible ($H_s$) heat fluxes anomaly composites, also shown the existence of antagonistic patterns in different stages of the SAMS for the groups of moist and dry soil. Specifically, anomalies of $H_l$ at the surface tend to be anomalously positive in the development quarter of the rainy season of SAMS, while their values decrease below the climatological average in the maturity quarter over the WCB under moist soil conditions. In contrast, for drier soils, an antagonistic pattern to the moist soil group is verified on the WCB. This implies that negative anomalies of $H_l$ at the surface are observed in the study area during the development quarter of the rainy season of SAMS, while values higher than the climatology are only identified in the subsequent quarter corresponding to the maturity stage of SAMS. This suggests that the extra energy for injecting more water vapor into the atmosphere occurs later in the dry soil groups compared to the moist soil group.

Regarding $H_s$ composites, it was observed that significant positive anomalies are usually higher over the study region during all stages of the rainy season of the SAMS for the dry soil group, while negative anomalies were observed for this variable in the moist soil group.

As for the intermediate soil, in general, no significant anomalies were observed for $H_l$ and $H_s$ at the surface, except for the weakening quarter when significant positive anomalies of $H_s$ tend to occur over the WCB. This is because, during the demise of the SAMS, the soil moisture and surface evaporation decrease due to the reduction in evaporation and the associated latent heat flux, resulting in a higher proportion of thermal energy being transferred as sensible heat from the soil surface, together with the gradual decrease in precipitation.





A contrasting pattern was verified for $H_s$ and $H_l$ in the neighborhood of 30° S and 65° W. There is a decrease in $H_s$ (increase in $H_l$) anomalies with wet soil conditions and an increase in $H_s$ (decrease in $H_l$) anomalies with dry soil conditions across

all stages of SAMS evolution. Similarly, this same area also exhibited anomalous behavior throughout the entire SAMS rainy period during for another variables. In the years characterized by wet soil conditions positive rainfall and surface/subsurface soil moisture anomalies, and negative $T2m$ and $PBLH$ anomalies were verified. Already in dry soil conditions, negative rainfall and surface/subsurface soil moisture anomalies, and positive $T2m$ and $PBLH$ anomalies were more common.

An analysis of the correlation between many hydrometeorological variables and rainfall over WCB during the rainy season

of SAMS between 1991 and 2021 revealed a significant positive correlation between rainfall and the surface soil moisture content ($SMCI_1$) and $B_0$. This indicates that moist soil contributes to the supply of moisture to the atmosphere, serving as an additional source of moisture for cloud formation, ultimately leading to rainfall over the study area. Other variables, such as $D_{ad}$ and $H_s$, also showed significant correlations with rainfall. Specifically, there was a positive correlation between the duration of active rainy days and WCB average precipitation during the SAMS rainy period, as well as a negative correlation

between $H_s$ and rainfall.

The significant negative correlation between $H_l$ and rainfall is due to the fact that relatively drier periods (which mainly include the SAMS rainy season with intermediate surface soil) experience greater solar radiation reaching the surface than the rainy season with wet surface soil, due the shorter durations of SAMS active days (less cloudiness). As a consequence, higher rates of evaporation ($EVP$) occur during these relatively drier periods, especially given that intermediate soils tend

to have relatively wetter conditions compared to drier soils and the low relative humidity values (not saturated environment with relative low actual vapor pressure) at the atmosphere contribute to increase the $EVP$. However, this alone does not fully explain the increase in the volume of rain over the WCB during the SAMS rainy season, because another dynamic processes in the atmosphere, like moisture advection processes, also contribute jointly to increasing the rainfall during this period.

The variables $D_{bd}$, $N_a$, $N_b$, $P_i$, $P_f$, $SMCI_4$, and $T2m$ do not exhibit significant direct correlations with rainfall over

WCB, and the indirect relationship between these variables and rainfall will be the subject of future research. Furthermore, for future work, there is still a need to improve the understanding of the relationship between rainfall in WCB region and different soil moisture conditions (dry, intermediate, and wet) across different areas of the South American continent and more detailed study using regional numerical weather prediction models will help to better understand how these different soil moisture scenarios affect different hydrometeorological variables during the development, maturity, and weakening SAMS

stages. These experiments will allow us to investigate how different remote soil moisture conditions in different areas of the South American continent interact with the precursor mechanisms of rainfall during the SAMS rainy period.

*Code availability.* The QGIS 3.16 Hannover software (available at: https://www.qgis.org/pt_BR/site/forusers/download.html) was used to create the geolocation map of the study area. The hydrometeorological maps utilized in this study were developed using Python version 3.7.10 (available at: https://anaconda.org/anaconda/python/files?version=3.7.10&page=0). The soil moisture data was grouped using the Ck-



means.1d.dp package (available at: https://cran.r-project.org/web/packages/Ckmeans.1d.dp/vignettes/Ckmeans.1d.dp.html) in Rstudio version 4.2.1 (available at: https://cran.r-project.org/bin/macosx/).

*Data availability.* Bathymetry-topography data were obtained from http://dss.ucar.edu/datasets/ds759.3. ERA5 daily and monthly reanalysis data were obtained from https://cds.climate.copernicus.eu/.

*Author contributions.* João Pedro Nobre produced all figures. All coauthors contributed to the analyses and to drafting this paper.

*Competing interests.* The contact author has declared that neither they nor their co-authors have any competing interests.

*Acknowledgements.* The present article contains results from the PhD thesis of the first author. CAPES (Coordenação de Aperfeiçoamento de Pessoal de Nível Superior-Brasil (CAPES)-Finance Code 001) and CNPq (Conselho Nacional de Desenvolvimento Científico e Tecnológico, doctoral scolarship: 140160/2021-3) are acknowledged for the finantial support.



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





**Table 1.** The average of surface and subsurface hydrometeorological variables ($B_o$, $H_l$, $H_s$, $PBLH$ $SMCI_1$, $SMCI_4$, and $T2m$), onset ($P_i$) and demise ($P_f$) pentads of the rainy season, number and duration of active ($N_a$ and $D_{ad}$, respectively)/break ($N_b$ and $D_{bd}$, respectively) periods in the rainy seasons of SAMS between 1991-2021, over WCB. The marks $\triangledown$, $\lozenge$ and $\triangle$, in column Wet Season, correspond respectively to dry, intermediate and wet soil condition.

| Wet Season | $B_o$ (-) | $D_{ad}$ (days) | $D_{bd}$ (days) | $H_l$ (W.m$^{-2}$) | $H_s$ (W.m$^{-2}$) | $N_a$ | $N_b$ | $PBLH$ (m) | $P_i$ | $P_f$ | Rain (mm.day$^{-1}$) | $SMCI_1$ (%) | $SMCI_4$ (%) | $T2m$ (°C) |
|---|---|---|---|---|---|---|---|---|---|---|---|---|---|---|
| 1991/1992$^\triangle$ | 0.26 | 10 | 9 | 114.20 | 29.29 | 6 | 1 | 467.58 | 54 | 18 | 7.55 | 80.24 | 62.53 | 25.30 |
| 1992/1993$^\triangle$ | 0.26 | 10 | 8 | 111.16 | 28.76 | 7 | 2 | 429.78 | 47 | 13 | 6.96 | 78.71 | 73.20 | 25.04 |
| 1993/1994$^\triangle$ | 0.28 | 17 | 5 | 112.53 | 31.44 | 4 | 1 | 465.19 | 59 | 17 | 7.89 | 79.90 | 58.56 | 25.71 |
| 1994/1995$^\triangle$ | 0.25 | 17 | 7 | 113.72 | 28.15 | 5 | 1 | 485.46 | 59 | 16 | 8.88 | 82.99 | 64.45 | 25.60 |
| 1995/1996$^\triangle$ | 0.27 | 9 | 0 | 117.65 | 31.22 | 6 | 0 | 497.16 | 60 | 16 | 7.30 | 81.42 | 59.09 | 25.42 |
| 1996/1997$^\triangle$ | 0.27 | 10 | 0 | 112.54 | 30.38 | 9 | 0 | 460.25 | 53 | 17 | 7.67 | 79.95 | 62.13 | 25.42 |
| 1997/1998$^\lozenge$ | 0.27 | 7 | 7 | 121.75 | 33.09 | 8 | 1 | 510.59 | 58 | 13 | 7.09 | 76.66 | 47.24 | 26.58 |
| 1998/1999$^\lozenge$ | 0.28 | 11 | 5 | 117.75 | 33.36 | 5 | 1 | 482.60 | 54 | 12 | 6.88 | 77.26 | 51.53 | 25.86 |
| 1999/2000$^\lozenge$ | 0.30 | 9 | 0 | 112.64 | 33.80 | 7 | 0 | 506.09 | 56 | 13 | 7.35 | 75.14 | 47.90 | 25.70 |
| 2000/2001$^\triangle$ | 0.27 | 10 | 13 | 120.82 | 32.91 | 5 | 1 | 495.95 | 57 | 14 | 6.61 | 77.24 | 53.54 | 25.77 |
| 2001/2002$^\triangle$ | 0.26 | 12 | 7 | 113.65 | 30.11 | 6 | 1 | 504.71 | 55 | 10 | 8.10 | 79.55 | 49.71 | 25.72 |
| 2002/2003$^\lozenge$ | 0.28 | 8 | 6 | 118.34 | 32.54 | 6 | 1 | 526.27 | 64 | 16 | 7.10 | 78.11 | 41.22 | 26.10 |
| 2003/2004$^\lozenge$ | 0.27 | 16 | 0 | 115.50 | 31.15 | 3 | 0 | 503.91 | 59 | 9 | 7.68 | 77.28 | 35.47 | 25.89 |
| 2004/2005$^\triangledown$ | 0.29 | 7 | 0 | 117.01 | 34.03 | 7 | 0 | 522.45 | 57 | 14 | 6.79 | 73.54 | 33.57 | 26.08 |
| 2005/2006$^\lozenge$ | 0.27 | 13 | 0 | 114.26 | 30.53 | 6 | 0 | 505.22 | 58 | 17 | 7.69 | 77.49 | 48.03 | 25.84 |
| 2006/2007$^\lozenge$ | 0.27 | 12 | 10 | 110.35 | 29.94 | 5 | 1 | 494.07 | 51 | 9 | 7.54 | 74.90 | 45.95 | 26.21 |
| 2007/2008$^\triangle$ | 0.27 | 11 | 0 | 115.17 | 31.04 | 6 | 0 | 513.59 | 57 | 13 | 8.18 | 79.57 | 44.80 | 25.64 |
| 2008/2009$^\triangledown$ | 0.29 | 7 | 8 | 114.91 | 33.49 | 8 | 1 | 524.41 | 57 | 16 | 7.22 | 73.12 | 48.50 | 26.10 |
| 2009/2010$^\lozenge$ | 0.28 | 10 | 5 | 113.48 | 31.91 | 6 | 1 | 512.76 | 55 | 12 | 7.81 | 76.02 | 50.18 | 26.18 |
| 2010/2011$^\lozenge$ | 0.30 | 12 | 6 | 109.56 | 32.61 | 6 | 1 | 530.75 | 56 | 16 | 8.02 | 76.88 | 44.10 | 25.74 |
| 2011/2012$^\triangledown$ | 0.31 | 10 | 6 | 111.89 | 34.40 | 7 | 1 | 524.89 | 54 | 14 | 7.23 | 72.08 | 48.38 | 25.68 |
| 2012/2013$^\lozenge$ | 0.28 | 11 | 0 | 114.08 | 31.93 | 6 | 0 | 531.86 | 59 | 17 | 7.69 | 76.81 | 43.39 | 26.14 |
| 2013/2014$^\triangledown$ | 0.31 | 10 | 9 | 110.70 | 33.80 | 6 | 1 | 500.55 | 53 | 17 | 7.18 | 73.07 | 45.22 | 25.69 |
| 2014/2015$^\lozenge$ | 0.28 | 7 | 10 | 115.20 | 31.75 | 6 | 1 | 500.98 | 57 | 21 | 6.98 | 75.64 | 55.40 | 25.94 |
| 2015/2016$^\lozenge$ | 0.27 | 11 | 0 | 122.14 | 32.59 | 3 | 0 | 544.75 | 60 | 12 | 7.35 | 74.89 | 42.32 | 26.71 |
| 2016/2017$^\lozenge$ | 0.27 | 13 | 5 | 117.97 | 32.26 | 4 | 1 | 489.06 | 55 | 16 | 7.35 | 76.95 | 45.72 | 25.96 |
| 2017/2018$^\triangle$ | 0.25 | 8 | 0 | 119.27 | 29.88 | 7 | 0 | 519.77 | 59 | 17 | 7.75 | 80.80 | 53.91 | 25.94 |
| 2018/2019$^\triangledown$ | 0.28 | 7 | 0 | 118.40 | 33.58 | 7 | 0 | 483.76 | 53 | 15 | 6.47 | 72.25 | 50.52 | 26.23 |
| 2019/2020$^\triangledown$ | 0.31 | 8 | 0 | 118.71 | 36.36 | 4 | 0 | 548.86 | 60 | 15 | 6.27 | 70.22 | 28.28 | 26.38 |
| 2020/2021$^\lozenge$ | 0.28 | 8 | 5 | 118.99 | 33.84 | 3 | 1 | 593.94 | 70 | 11 | 6.68 | 74.96 | 9.10 | 25.84 |
| **Mean** | 0.28 | 10.40 | 4.35 | 115.48 | 32.00 | 5.80 | 0.63 | 505.91 | 56.87 | 14.53 | 7.38 | 76.79 | 48.13 | 25.88 |
| $\sigma$ | 0.02 | 2.81 | 4.00 | 3.43 | 1.86 | 1.49 | 0.56 | 30.64 | 4.12 | 2.80 | 0.56 | 3.04 | 11.90 | 0.36 |