# Peer review of "Effects of soil moisture and surface heat fluxes on the South American Monsoon System over West-Central Brazil: an observational study"

_Hydrology and Earth System Sciences, 2023_

## Referee Comment (RC2)

This study analyzed the hydrometeorological characteristics during the South American Monsoon System (SAMS) over West-Central Brazil (WCB). At present, I think that the results of this study are insufficient to support the conclusions, and there is still a lack of analysis of the physical mechanisms of land-atmosphere coupling. I am not sure whether this work rises to the high level of quality and significant impact expected by HESS. Thus, I recommend rejecting this manuscript. My comments are provided below.

1. Section 3.2.1: I think there may be a problem with the logic of this article. I do not agree that using composite analysis alone can prove that the soil moisture is a key variable in defining spatiotemporal precipitation patterns. On the contrary, perhaps the precipitation is an important factor in determining the spatiotemporal of soil moisture. Soil moisture may have an impact on precipitation through land-atmosphere coupling, but this study does not provide any sufficient evidence.

2. The results section of this study cited many references to demonstrate the relevant results. What are your innovative points compared to these studies?

3. Line 188: Only 6 years were identified as wet years in this study, and the composite analysis may have significant uncertainty due to the limited samples. ERA5 can provide long-term reanalysis data, and perhaps you can carry out the study in longer years.

4. Lines 142-145、171-180: The spatiotemporal changes in precipitation and radiation should be provided to validate relevant statements.

5. Line 237-241: The relationship between evapotranspiration and precipitation in this study is ambiguous. You think the increase in evapotranspiration is an important reason for the increase in precipitation, but there is a significant negative correlation between precipitation and evapotranspiration over WCB (Figure 7).

 Minor Comments

6. Table 1 is very non intuitive.

7. Please indicate the supplement Figures in the manuscript.

8. Figure 7 contains a lot of information, but not all of them are meaningful. Perhaps a more concise and clear expression can be used. In addition, what's the mean of "0.1(.), 1()"? please check the figure caption.

9. Figure 11: Please reverse the colorbar.

10. Lines 349-351: please check this sentence.

11. Line 357: "then" should be changed into "than", please check it.

12. Line 382: We generally do not use percentile form to represent the correlation coefficient.

13. Please explain the meanings of $D_{ad}$, $D_{bd}$, $N_a$, $N_b$, $P_i$, $P_f$ and $SMCI_4$ when they first appear in the manuscript.

14. Lines 439-442: Please check this sentence.

---

## Author Comment (AC3)

[Figure]

**Figure S1.** Climatology of 1991-2021 for volumetric soil moisture at 0-7 cm (column a-g), in $m^3.m^{-3}$, and average daily precipitation (column b-h), in $mm.day^{-1}$, for the quarters of September-October-November (SON, line a-b), December-January-February (DJF, line c-d), March-April-May (MAM, line e-f) and June-July-August (JJA, g-h)

[Figure]

**Figure S2.** Climatology of 1991-2021 for air temperature at 2 meters (column a-j), in °C, and sensible (column b-k) and latent (column c-l) heat fluxes at the surface, in $W.m^{-2}$, for the September-October-November (SON, line a-C), December-January-February (DJF, line d-F), March-April-May (MAM, line g-i) and June-July-August (JJA, j-l) quarters.

[Figure]

**Figure S3.** TCI (W.m$^{-2}$) over the development (SON, column a-g), maturity (DJF, column b-h), and weakening (MAM, column c-i) quarters of the SMAS rainy season for three soil moisture conditions; wet (line a-c), intermediate (line d-f), and dry (line g-i). The hatched area shows statistical significance when the p-value was less than 0.05.

[Figure]

**Figure S4.** ACI (mm.day$^{-1}$) over the development (SON, column a-g), maturity (DJF, column b-h), and weakening (MAM, column c-i) quarters of the SMAS rainy season for three soil moisture conditions; wet (line a-c), intermediate (line d-f), and dry (line g-i). The hatched area shows statistical significance when the p-value was less than 0.05.

[Figure]

**Figure S5.** TF (W.m$^{-2}$.mm.day$^{-1}$) over the development (SON, column a-g), maturity (DJF, column b-h), and weakening (MAM, column c-i) quarters of the SMAS rainy season for three soil moisture conditions; wet (line a-c), intermediate (line d-f), and dry (line g-i). The hatched area shows statistical significance when the p-value was less than 0.05.